# Anti-Virulence Properties of *Coridothymus capitatus* Essential Oil against *Pseudomonas aeruginosa* Clinical Isolates from Cystic Fibrosis Patients

**DOI:** 10.3390/microorganisms9112257

**Published:** 2021-10-29

**Authors:** Gianluca Vrenna, Marco Artini, Rino Ragno, Michela Relucenti, Ersilia Vita Fiscarelli, Vanessa Tuccio Guarna Assanti, Rosanna Papa, Laura Selan

**Affiliations:** 1Department of Public Health and Infectious Diseases, Sapienza University, p.le Aldo Moro 5, 00185 Rome, Italy; gianluca.vrenna@uniroma1.it (G.V.); marco.artini@uniroma1.it (M.A.); 2Department of Drug Chemistry and Technology, Sapienza University, p.le Aldo Moro 5, 00185 Rome, Italy; rino.ragno@uniroma1.it; 3Rome Center for Molecular Design, Department of Drug Chemistry and Technology, Sapienza University, p.le Aldo Moro 5, 00185 Rome, Italy; 4Department of Anatomy, Histology, Forensic Medicine and Orthopaedics, Sapienza University of Rome, via Alfonso Borelli 50, 00161 Rome, Italy; michela.relucenti@uniroma1.it; 5Unit Cystic Fibrosis Diagnostic Microbiology and Immunology Diagnostics, Diagnostic Medicine and Laboratory Department, Bambino Gesù Children’s Hospital, 00165 Rome, Italy; evita.fiscarelli@opbg.net (E.V.F.); vanessa.tuccio@opbg.net (V.T.G.A.)

**Keywords:** *Pseudomonas aeruginosa*, cystic fibrosis, essential oil, antivirulence, SEM, motility, biofilm, pyocyanin, *Coridothymus capitatus* essential oil

## Abstract

*Pseudomonas aeruginosa* is an opportunistic pathogen responsible for nosocomial infections, and is often involved in airway infections of cystic fibrosis (CF) patients. *P. aeruginosa* virulence is related to its ability to form biofilm, trigger different types of motilities, and produce toxins (for example, bacterial pigments). In this scenario, essential oils (EOs) have gained notoriety for their role in phenotype modulation, including virulence modulation. Among different EOs previously analyzed, herein we investigated the activity of *Coridothymus capitatus* EO (CCEO) against specific virulence factors produced by *P. aeruginosa* isolated from CF patients. CCEO showed inhibition of new biofilm formation and reduction in mature biofilm in about half of the tested strains. On selected strains, SEM analysis provided interesting information regarding CCEO action in a pre-adhesion assay. CCEO treatment showed a dramatic modification of the extracellular matrix (ECM) structure. Our results clearly showed a drastic reduction in pyocyanin production (between 84% and 100%) for all tested strains in the presence of CCEO. Finally, CCEO was also able to strongly affect *P. aeruginosa* swarming and swimming motility for almost all tested strains. In consideration of the novel results obtained on clinical strains isolated from CF patients, CCEO may be a potential candidate to limit *P. aeruginosa* virulence.

## 1. Introduction

The opportunistic pathogen *Pseudomonas aeruginosa* is one of the main bacteria responsible for nosocomial infections in humans, with a high incidence of infection occurring in immunocompromised patients and in patients with cystic fibrosis (CF) [1,2]. CF is a genetic disorder caused by the presence of mutations in the *CFTR* gene encoding for a protein called cystic fibrosis transmembrane conductance regulator (CFTR), a cell-surface localized chloride channel that regulates absorption and secretion of salt and water across epithelia [3]. The presence of mutations has been reported to alter the transport across the cellular membrane, especially in the airways [4]. Lungs with a defective CFTR protein are characterized by a thick and sticky mucus that clogs the airways and traps bacteria and fungi, leading to inflammation, and recurrent and chronic infections, in turn leading to respiratory failure and other complications [5].

The airways of CF individuals are first colonized by *Haemophilus influenzae* and *Staphylococcus aureus* and then, until to the end of the patient’s life, by the opportunistic pathogen *P. aeruginosa* [6,7]. Elevated osmotic stress, very high concentrations of antibiotics, reduced nutrient availability, and intermicrobial competition render the lung environment extremely aggressive, forcing *P. aeruginosa* to adapt for the survival [8]. For these reasons, *P. aeruginosa* acquires increasing levels of antimicrobial resistance (AMR) in response to treatments. AMR, together with the extensive number of virulence factors, renders *P. aeruginosa* an extremely audacious and fearsome pathogen [9].

During long-term lung infection in CF patients, *P. aeruginosa* strains develop mutations leading to clonal expansion. This microevolution is believed to be correlated with a reduced virulence. Various phenotypic traits have been described in *P. aeruginosa* strains derived by chronic infections, which are absent in environmental *P. aeruginosa* strains, including loss of motility, acquisition of mucoidy, and antibiotic resistance [10].

Indeed, in the literature it is reported that microevolution within CF lungs leads to the selection of *P. aeruginosa* strains with altered but not reduced virulence. This evolution does not merely reflect the natural course of infection, because it is the result of the interaction between pathogen, host, and treatment. Furthermore, the genetic adaptation of *P. aeruginosa* can be distinct in differentially predisposed hosts [11].

The virulence of each *P. aeruginosa* isolate is strongly related to the capability to form biofilm, trigger different types of cell/colonial motility, and produce toxins (for example, bacterial pigments) [12]. *P. aeruginosa* has at least three types of motilities, including swarming, swimming, and twitching [13]. Different types of motilities play a pivotal role in bacterial colonization of surfaces and in the formation of biofilms [14]. Biofilm formation is a dynamic process that allows free-living bacteria to be protected from and resistant to drugs, and to host immune attacks. In a biofilm, different bacterial communities are incorporated in a self-produced exopolysaccharide matrix (EPS) that protects bacteria from antibiotics, ensuring their survival and complicating their eradication [15]. In addition, pyocyanin production is a further virulence factor, which induces oxidative stress directly proportional to the severity of the disease [16]. The redox activity of pyocyanin is important in the establishment of infections, interfering in numerous cellular processes of host [17]. Expression of these virulence factors is mainly controlled by quorum sensing (QS), an intercellular communication system that bacteria adopt to monitor the population density (via signaling molecules and receptors) and adapt the gene expression to the constantly changing needs [18,19,20].

The inhibition of the QS system has been considered as a novel strategy for the development of antipathogenic agents [21,22,23,24,25,26]. Quorum sensing inhibitors (QSIs) acting on specific bacterial virulence programs do not affect bacterial vitality; thus, they should not lead to the development of the resistance observed for antibiotics [27]. In particular, phenolic compounds, terpens and terpinenes, which are widely distributed in plants and especially concentrated in essential oils (EO), exert many important physiological functions. Among these, many were reported to reduce QS controlled phenotypes in bacteria such as *P. aeruginosa*, *E. coli*, and *S. aureus* [28,29,30,31]. Furthermore, EO have also been reported to display anti-virulence activity in many studies [31,32]. Taking into account these features, EOs can be considered promising candidates for both therapy and disinfection of medical devices. In our previous reports, many EOs were investigated for their abilities to modulate bacterial biofilm production of different clinical and reference strains belonging to *Pseudomonas* spp. and *Staphylococcus* spp. [33,34,35,36].

In this study we investigated the anti-virulence properties of EO obtained from *Corydothymus capitatus*, a perennial aromatic shrub belonging to the *Lamiaceae* family, and native to the Mediterranean basin. The most abundant components in the EO used in this work are carvacrol, β-caryophyllene, and p-cymene, which are well known for their antifungal, antimicrobial, phytotoxic, and insecticidal activities [37]. Our previously published studies showed an antimicrobial activity for this EO against *S. aureus* strains isolated from CF patients [35,36], according to data reported in the literature [38,39]. Herein we deeply investigated the activity of *Coridothymus capitatus* EO (CCEO) against specific virulence factors produced by *P. aeruginosa* isolated from CF patients, such as biofilm formation and accumulation, pyocyanin production, and swimming and swarming motility. CCEO is known to have specific biological properties and, due to its chemical profile rich in phenolic compounds, terpens and terpinenes, it is able to modulate phenotype in bacteria [38]. The effect of CCEO was also investigated by Scanning Electron Microscopy (SEM) analysis on selected *P. aeruginosa* reference and clinical strains.

## 2. Materials and Methods

### 2.1. Ethics Approval and Informed Consent

The approval for this research was granted by the Ethics Committee of Children’s Hospital and Institute Research Bambino Gesù in Rome, Italy (No 1437_OPBG_2017 of July 2017), and it was performed according to the principles of the Helsinki Declaration. Informed consent was obtained from all subjects aged 18 years and older and from parents of all subjects under 18 years of age prior to enrolment.

### 2.2. Bacterial Strains and Growth Conditions

In this study, eleven *P. aeruginosa* strains, isolated from respiratory specimens of 7 CF patients in follow-up at Pediatric Hospital Bambino Gesù (OPBG) of Rome, Italy, were used. Phenotypic and genotypic characteristics of the bacterial strains are summarized in Appendix A. *P. aeruginosa* PA14 was used as the reference strain because it is recognized as an excellent model for studies on pathogenesis and biofilm formation [38]. Microbiological cultures were performed according to approved guidelines, using selective media, and manual and automatic systems (API20NE, Vitek2, MALDI-TOF mass spectrometry); the unambiguous identification was performed by sequencing 16S rRNA. The strains, selected from a bacterial collection including about 10,000 isolates from CF patients attending the OPBG Center, showed different phenotypic and biochemical features to better represent the complexity of CF lung microbiota population. Susceptibility testing to carbapenems (imipenem, meropenem), piperacillin/tazobactam, aminoglicosides (tobramycin, amikacin), quinolones (ciprofloxacin, levofloxacin), monobactam (aztreonam), and cephalosporins (ceftazidime, cefepime) was carried out by Minimum Inhibitory Concentration (MIC) determined by an E-test on Mueller Hinton (MH) agar plates, according to EUCAST criteria. The colistin MIC values were evaluated by Broth Microdilution (ComASP Colistin Liofilchem, Italy).

Bacteria were grown in Brain Heart Infusion broth (BHI, Oxoid, Basingstoke, UK). Planktonic cultures were grown in flasks under vigorous agitation (180 rpm) and biofilm formation was performed in static conditions, at 37 °C.

### 2.3. CCEO

The CCEO was purchased from Farmalabor srl (Assago, Italy). As reported by the manufacturer, CCEO was obtained by steam distillation of whole flowering plants obtained from controlled organic crops. CCEO was solubilized by adding dimethyl sulfoxide (DMSO) in a ratio of 1:1, to generate a mother stock solution of 50% *v/v*. Chemical analyses of CCEO were performed as reported in Papa et al. (2020) [36]. Chemical composition is reported in Table 1.

In this study, CCEO was selected to analyze its possible activity against some virulence features of a diversified spectrum of *P. aeruginosa* isolates. In particular, as previously reported, *P. aeruginosa* isolates were selected on the basis of their ability to form biofilm, to produce pyocyanin, and to swarm and swim. Preliminarily, the antimicrobial activity (MIC) of CCEO was determined using the microdilution method. It showed bactericidal activity only against strain 40P at a higher tested concentration (5% *v/v*).

### 2.4. Determination of Minimal Inhibitory Concentrations

The MIC of CCEO was determined as the lowest concentration at which observable bacterial growth was inhibited. MICs were determined according to the guidelines of Clinical Laboratory Standards Institute (CLSI, 2018) [40]. EO solubilized in DMSO was used, starting from a concentration of 5% *v/v*. As the control, bacterial cultures were performed in DMSO at a concentration of 5% *v/v*. Appropriate dilution (10^6^ CFU/mL) of bacterial cultures in exponential phase were used. All experiments were performed in quadruplicate.

### 2.5. Biofilm Formation Assays

The effect of CCEO was investigated at two different timepoints during biofilm development; it was added to the medium at the beginning of the cultivation (0 h, pre-adhesion period), and after biofilm formation (24 h of bacterial culture). The CCEO was applied at a concentration of 1% *v/v* solubilized in DMSO (final concentration 1%, *v/v*), chosen on the basis of a previous report [36]. As the control, bacteria were cultured in presence of 1% DMSO.

#### 2.5.1. Pre-Adhesion Period

Biofilm production was quantified based on microtiter plate biofilm assay (MTP) as previously reported [36]. Briefly, the wells of a sterile 96 well flat-bottomed polystyrene plate were filled with BHI containing a 1/100 dilution of overnight bacterial cultures (about 0.5 OD 600 nm). As the control, the first row contained the untreated bacterial cells in BHI broth with 1% *v/v* DMSO. In the second row the same bacterial culture was added with EO at a concentration of 1% *v/v*. The plates were aerobically incubated for 18 h at 37 °C. After the incubation, the well content was aspirated, washed three times with double-distilled water to remove planktonic cells, and the plates were dried in an inverted position. For the quantification of biofilm formation, each well was stained with 100 µL of 0.1% crystal violet and incubated for 15 min at room temperature, rinsed twice with double-distilled water, and thoroughly dried. The remaining dye attached to the adherent cells was solubilized with 20% (*v/v*) glacial acetic acid and 80% (*v/v*) ethanol. After 30 min of incubation at room temperature, the total biofilm biomass in each well was spectrophotometrically quantified at 590 nm. Each data point is composed of 4 independent experiments, and each experiment was replicated at least 6 times.

#### 2.5.2. Mature Biofilm

An assay on preformed biofilm was also performed. The wells of a sterile 96 well flat-bottomed polystyrene plate were filled with 100 µL of BHI medium containing 1/100 dilution of overnight bacterial culture. The plates were aerobically incubated for 24 h at 37 °C. Then, the contents of the plates were poured off and the wells were washed to remove the unattached bacteria, and 100 μL of the fresh medium containing or not containing 1% *v/v* of EO was added to each well. The inoculated plates prepared in this way were aerobically incubated for an additional 24 h (48 h in total) at 37 °C. After 24 h the plates were analyzed as previously described.

### 2.6. Pyocyanin Assay

Pyocyanin production was determined as described by Pejčić and coworkers [12] with modifications. Bacterial cells were inoculated in BHI broth with or without EO at 0.5% (*v/v*) and incubated for 72 h at 37 °C. As the control, the BHI was supplemented with 0.5% of DMSO. The cells were removed by centrifugation (10,000 rpm, 15 min) and the supernatant was used for pyocyanin extraction. Briefly, 2 mL of chloroform was added to 2 mL of the supernatant. The solution was mixed for 2 min by inversion and then decanted for 15 min to allow the separation of organic phase to aqueous one. The lower layer containing pyocyanin was transferred to a tube containing 2 mL of 0.2 M HCl. The resulting solution was mixed and decanted to allow the separation of the two phases. Then, the pink colored upper layer was separated and pyocyanin was subsequently spectrophotometrically quantified at 520 nm.

### 2.7. Motility Assays

#### 2.7.1. Swarming Assay

The swarming assay was performed as previously published by Yang and coworkers [41], with some modifications. Briefly, the EO dissolved in DMSO was added to molten swarming agar at a concentration of 0.5 % (*v/v*). Swarming agar was prepared as follows: 0.8% Nutrient Broth (Oxoid, Basingstoke, UK), 0.5% D-(+)-glucose (Sigma, Steinheim, Germany), and 0.5% agarose (Invitrogen, Paisley, UK). The culture was then dispensed onto Petri dishes after gentle mixing. Once the culture was solidified, 2 μL of each overnight *P. aeruginosa* culture was inoculated in the center of the agar and then incubated at 37 °C for 24 h. We used DMSO as the control (1% *v/v*). After the incubation period, diameters of the growth zones were measured. Anti-QS properties were identified by the reduction in swarming motility.

#### 2.7.2. Swimming Assay

The swimming assay was conducted according to previous research [41], with some modifications. The procedures were the same as those of the swarming assay, except for the swimming agar composition, which consisted of 1.0% peptone (Oxoid, Basingstoke, UK), 0.5% sodium chloride (Sigma, Steinheim, Germany), and 0.3% Bacto-Agar (BD, Le Pont de Claix, France). After the incubation period, diameters of the growth zones were measured.

### 2.8. SEM Analysis

For SEM analysis, bacteria were grown as follows: briefly, 1/100 dilution of overnight bacterial cultures was transferred in tubes containing SEM stubs (aluminum, 12.5 mm diameter, 6 mm pin) and incubated for 18 h at 37 °C in static condition allowing biofilm production, in BHI and in the presence or absence of 0.5% (*v/v*) EO. After the growth, SEM stubs were washed in 0.1 M phosphate buffer pH 7.4 (PB) and fixed in 2.5% glutaraldehyde in 0.1 M PB buffer. Samples were washed overnight in PB and postfixed with a mixture of 2% OsO_4_ and 0.2% Ruthenium Red, for 1 h at room temperature [42,43]. Samples were then washed for 30 min with H_2_O. The excess water was dried carefully with filter paper, then the samples were mounted on the specimen holder and observed in a Hitachi SU3500 microscope (Hitachi, Japan), at variable pressure conditions of 5kV and 30Pa.

Three-dimensional reconstruction was undertaken by Hitachi Map 3D Software (v.8.2., Digital surf, Besançon, France) [44]. A single image reconstruction procedure was used and, from the 3D reconstructed image, a representative area was extracted. The surface topography of the extracted area is shown in false colors.

### 2.9. Statistical Analysis of Biological Evaluation

Data reported were statistically validated using a Student’s t-test comparing mean absorbance of treated and untreated samples. The significance of differences between mean absorbance values was calculated using a two-tailed Student’s *t*-test. A *p*-value of <0.05 was considered significant.

## 3. Results

### 3.1. Phenotypic Characterization of Clinical and PA14 Strains

The selected *P. aeruginosa* isolates were characterized by the presence of specific virulence factors, such as biofilm formation, pyocyanin production, and swarming and swimming motility. The evaluation of these features in the studied strains is reported in Table 2.

### 3.2. Effect of EO on Biofilm Formation

The results of the CCEO anti-biofilm activity after application at the beginning of the cultivation period (0 h) are presented in Figure 1, panel A. The results are expressed as the percentage of biofilm formed in the presence of CCEO compared to the untreated sample. CCEO showed anti-adhesion activity on about half of the tested strains, inhibiting biofilm formation in a variable percentage (mostly from 60 to 40%). The strongest inhibition (95.7%) was achieved against the isolate 29P, whereas the strain 26P was stimulated towards growth, achieving 166% of biofilm formation compared with the control. Furthermore, the CCEO also showed inhibitory activity against the reference strain PA14, which is a highly virulent strain characterized by a hyperbiofilm formation due to a specific mutation in the *retS* gene [38].

The ability of the CCEO to degrade mature biofilms is reported in Figure 1, panel B. After addition of the CCEO, in five of 12 strains a biofilm reduction ranging from 27 to 52% was observed. However, the CCEO seemed to favor biofilm accumulation on some strains, as already highlighted for other EOs (for example mandarin, patchouli, and cedar fruit EOs) [36].

### 3.3. Effect of CCEO on Pyocyanin Production

All tested isolates in the current study were producers of pyocyanin (Table 2). The present study investigated the effects of the CCEO on the ability of the bacteria to synthesize and secrete this virulence-associated pigment. The results on the pyocyanin production in the presence of CCEO are presented in Table 3. The EO strongly reduced pyocyanin production, thoroughly inhibiting the production of the pigment in 8 of 12 strains.

### 3.4. Effect of CCEO on Motility

The ability of CCEO to reduce swarming and swimming motility patterns in clinical *P. aeruginosa* isolates was also assayed. The type of motility pattern and the influence on it by CCEO is represented in Figure 2 and Figure 3.

Swarming is defined as a fast, coordinated movement of bacteria on a semi-solid surface. CCEO was tested at two different concentrations, as reported in Figure 2. CCEO (at both concentrations) reduced swarming zones in all tested strains (Figure 2). In such cases, bacteria were not macroscopically visible on agar plate, even though the concentrations used were not able to impair bacterial vitality. In such cases, motility was completely inhibited by the presence of CCEO, and only a few scattered colonies were visible (strains 30P, 32P, and 34P).

The swimming of bacteria in liquid environments is enabled by polar flagella movements. The presence of the CCEO at 1% and 0.5% *v/v* caused a strong reduction in swimming motility in all tested strains. This effect was not dose dependent; in fact, in such cases, it was more evident at lower concentrations of CCEO used.

### 3.5. SEM Analysis

#### 3.5.1. Evaluation of the Effect of CCEO on *P. aeruginosa* PA14

Ultrastructural observation of untreated *P. aeruginosa* PA14 samples showed the presence of an abundant extracellular matrix (ECM), whose surface appears very rough (Figure 4A). In the images at higher magnification (Figure 4B), the ECM of untreated *P. aeruginosa* shows a spongy appearance, derived by a network structure including larger meshes in some areas and smaller ones in others. The ECM three-dimensional structure of the same untreated strain shows a regularly branching trabecular system, perforated by a labyrinthic arrangement of narrow channels. (Figure 4C). The treatment of *P. aeruginosa* PA14 with CCEO induces dramatic changes of its morphology: lower magnification (Figure 4D) images show ECM with a smooth surface and a diffuse, more solid aspect. Increasing the magnification (Figure 4E), the smooth surface reveals a fine granular texture where no channels or networks are visible. At high magnification (Figure 4F), the three-dimensional structure appears as a dense and irregular aggregate of globular units, perforated by low-depth pits and very narrow, irregular microchannels. Comparing image Figure 4B with Figure 4E, the disappearance of a regular network structure is clearly evident; this structural change is possibly due to a collapse of the trabecular structure into a compact mass, as is evident looking at the differences between images Figure 4C,F.

To further analyze and reveal the fine structural differences between the treated sample and the control, a three-dimensional reconstruction of the images reported in Figure 4C,F was carried out using the Hitachi Map 3D Software (v.8.2., Digital surf, Besançon, France). Figure 5A shows the three-dimensional reconstruction of the high-magnification image of the control sample; in Figure 5B, the circular area extracted from Figure 5A is magnified and represented in false colors that emphasize its large-mesh network structure. Figure 5C shows the three-dimensional reconstruction of the high-magnification image of the treated sample; Figure 5D shows the circular area extracted from Figure 5C, magnified and represented in the same false colors as in Figure 5B. In these figures it is easy to appreciate the absence of a mesh structure (no large blue areas, only scattered small points), whereas most of the surface consists of compact elevated areas (turquoise-green and yellow-red).

#### 3.5.2. Evaluation of the Effect of CCEO on *P. aeruginosa* 31P

Ultrastructural images of untreated *P. aeruginosa* 31P samples show that this strain produces abundant and dense ECM (Figure 6A), which appears compact in the bulk and raw on the surface due to the presence of fine globular aggregates (Figure 6B). At higher magnification, smaller aggregates are visible, which melt together and form larger aggregates (Figure 6C). The effect of CCEO treatment on *P. aeruginosa* 31P ECM is dramatic, even at low magnification (Figure 6D), where biofilm appears as a large meshwork, as if ECM was “digested”. At higher magnification (Figure 6E), the disruptive action of the CCEO is more evident: the walls of a single mesh appear to be formed by a short-branching trabecular system with a curled and irregular aspect. In the left part of the picture, the trabecular system is thicker and less disrupted than in the right part. The image at highest magnification (Figure 6F) shows a still undisrupted thick trabeculae in the left; in the right, thin and collapsing trabeculae with very irregular surfaces are visible.

Moreover, for this strain a three-dimensional reconstruction of images reported in Figure 6C,F was carried out using the Hitachi Mountains Map Advanced Software (v.8.2., Digital surf, Besançon, France), to better characterize the structural differences between the treated sample and the control. Results are illustrated in Figure 7. Figure 7A shows the three-dimensional reconstruction of the high-magnification image of the control sample; the circular area extracted from Figure 7A is magnified in Figure 7B and represented in false colors that emphasize the compact structure. The very few perforated areas are stained in blue, but most of the structure is compact and elevated, because yellow-red areas (highest areas) strongly prevail compared to turquoise-green areas (mildly elevated areas). Figure 7C shows the three-dimensional reconstruction of the high-magnification image of the treated sample. In Figure 7D, the circular area extracted from Figure 7C is magnified and represented in false colors, as in Figure 7B. Only a small area is occupied by an elevated structure (yellow-red), whereas most of the area comprises a middle-elevated structure (turquoise-green) perforated by large deep holes (blue areas).

## 4. Discussion

In this study, attention was focused on virulence factors phenotypically expressed in *P. aeruginosa* isolates from CF patients. Factors such as biofilm formation, pyocyanin production, and motility capability are required for full virulence, suggesting their universal role during infection. Mostly they are fundamental to allow the adaptation and the survival of bacteria in an extremely aggressive environment, such as the lung of CF patients [7]. In this study, *P. aeruginosa* PA14 was used as the reference strain because it displays higher virulence in most hosts compared to the model organism *P. aeruginosa* PAO1 [45]. Its virulence is linked to several factors, such as motility, quorum sensing, and biofilm formation. Previous reports showed that a specific mutation in a gene named *lads* results in deregulation of biofilm formation and enhanced production of the T3SS, leading to elevated cytotoxicity in this strain [46].

Chronic lung infections, such as those occurring in CF, require intensive use of antimicrobial drugs, which inevitably lead to the selection of antibiotic resistant bacterial strains. In order to overcome antibiotic resistance during the treatment of CF lung infections, new compounds should be identified. In practice, research is focused on alternative strategies aimed to disarm pathogens, without killing or damaging them, and therefore limiting the selective pressure that promotes the antibiotic resistance phenomenon. In this scenario, essential oils represent ideal candidates because they may act as anti-virulence or antipathogenic agents [12,14,16]. EOs are complex mixtures of different classes of organic compounds and have been empirically used in traditional medicine, including for the treatment of uncomplicated upper respiratory tract infections [47,48]. Some EOs include essentially terpenoids and phenolic compounds that are known to impair the virulence by interference with bacterial quorum sensing mechanisms [31]. Furthermore, EOs are multi-component mixtures, and the bacteria cannot become resistant to all of the components of the oils at the same time. For this reason they fail to develop effective resistances.

Previous studies showed the antimicrobial activity of several commercial essential oils against bacterial isolates from CF patients belonging to *P. aeruginosa* and *S. aureus* species [35]. The chemical composition of these EOs was previously reported [36]. Herein, we demonstrated that CCEO did not show antimicrobial activity at tested concentrations, but it strongly affected some specific virulence features. The effect of CCEO on biofilm formation was strongly strain dependent, similar to the results previously reported for several EOs [33,34,36]. This result was evident in all phases of biofilm formation: in pre-adhesion assay by adding EO from the beginning of the bacterial culture, and on mature biofilm by adding EO at 24 h after biofilm establishment. In both cases, we observed a positive effect on biofilm inhibition and/or disaggregation in at least 50% of the bacterial strains analyzed (Figure 1). Particularly of interest is the disaggregating activity shown by EO on mature biofilm; in fact, once a biofilm is established, it is very hard to disrupt it. It must be highlighted that the control (untreated) biofilms formed by 34P and 40P (Figure 1, panel B), grown after 48 h of incubation replacing the medium after 24 h, were definitely more abundant and presumably more structured and difficult to eradicate.

The images obtained by SEM provided interesting information regarding the action of CCEO on two of the 12 tested *P. aeruginosa* strains. Strain 31P was chosen for this analysis, together with reference strain PA14, because it was a strong biofilm producer (Table 2) and its biofilm formation was severely impaired by CCEO treatment (Figure 1A).

Even if major structural differences exist in the ECM structure of the two strains before treatment (untreated PA14 showed a spongy structure, whereas 31P was characterized by a compact structure), after CCEO treatment both strains showed a dramatic modification of the ECM structure. The strain with the spongy structure (PA14) has its meshwork architecture disrupted by EO action, and the collapse of its 3D structure results in a compact mass. On the contrary, EO breaks apart the compact ECM of the 31P strain. In both cases, CCEO acts as an ECM disintegrating agent: in the case of PA14 it destroys an already lax structure, causing it to collapse; in the case of 31P, CCEO breaks down a dense structure, causing its disintegration. Because the macroscopic data on biofilm growth obtained by the Christensen method unequivocally demonstrated a reduction in the whole biofilm mass, the two different patterns have to be interpreted as two different consequences of the dispersal and erosion of the ECM. The erosion of ECM in the rough structure of untreated PA14 produced a smooth surface and a diffuse, more solid aspect due to the collapse of the fine trabeculae peculiar of the untreated compact structure; on the contrary, in 31P the erosion induced a diffuse wormhole of the previous compact structure, as if it were digested. This interpretation was confirmed by the Hitachi Map 3D Software (v.8.2., Digital surf, Besançon, France) analysis, which clearly shows, in false color, the variation of surface topography of the samples, in terms of the increase or diminishing of deep areas and elevated areas.

In addition to biofilm, *P. aeruginosa* possesses other virulence factors. Pyocyanin, as an example, is used to establish infections and to promote virulence by interfering in numerous cellular functions in host cells due to its redox activity [17]. This green pigment, whose production is controlled by QS, is secreted by bacteria and allows them to grow preferentially in unfavorable environments, including by aiding the development of *P. aeruginosa* biofilm [49]. Pyocyanin also acts as an antimicrobial agent against several species of other bacteria and fungi. It is involved in the oxidative stress on lung epithelial cells, leading to lung damage, respiratory failure, and death [50]. Our results clearly showed a drastic reduction in pyocyanin production for all tested strains in the presence of CCEO, likely suggesting an interference with QS activation and regulation. The decrease in the production of pyocyanin is reported in literature for EOs and extracts obtained from several plants, such as basil [8], olive leaf [51], and cinnamon [32], with different efficiencies ranging from 22% to 97%.

Finally, CCEO was also able to strongly affect *P. aeruginosa* swarming and swimming motility for almost all tested strains. Also in this case, several reports describe the effect of natural extracts or EOs on the ability of bacteria to swarm or swim [12]. A plausible explanation for this result can be found by analyzing the chemical composition of CCEO, which is particularly rich in carvacrol (61%). It is well known that carvacrol reduces *lasR* gene expression and, consequently, inhibits LasI activity. As reported in the literature, the latter affects biofilm formation, pyocyanin production, and swarming motility [52,53].

## 5. Conclusions

In consideration of the novel results obtained from clinical strains isolated from CF patients, CCEO may be a candidate for use in procedures to disinfect patients’ used equipment. This application is also corroborated by the fact that EOs seem to possess a moderate toxicity on human cells in culture. Respiratory therapists pay significant attention to the treatment and education of patients with CF, particularly with regard to infection-control recommendations. The main routes of pathogen transmission are direct contact (patient-to-patient); indirect contact (contaminated objects); and droplet (droplets of liquid emitting from exhaled breath or produced by medical procedures such as aerosol administration) [54]. For this reason, the correct disinfection of respiratory care equipment, particularly the nebulizer, with proper solutions (i.e., EOs) may inhibit bacterial proliferations and enable biofilm reduction to reduce pathogen transmission via contaminated objects.

## Figures and Tables

**Figure 1 microorganisms-09-02257-f001:**
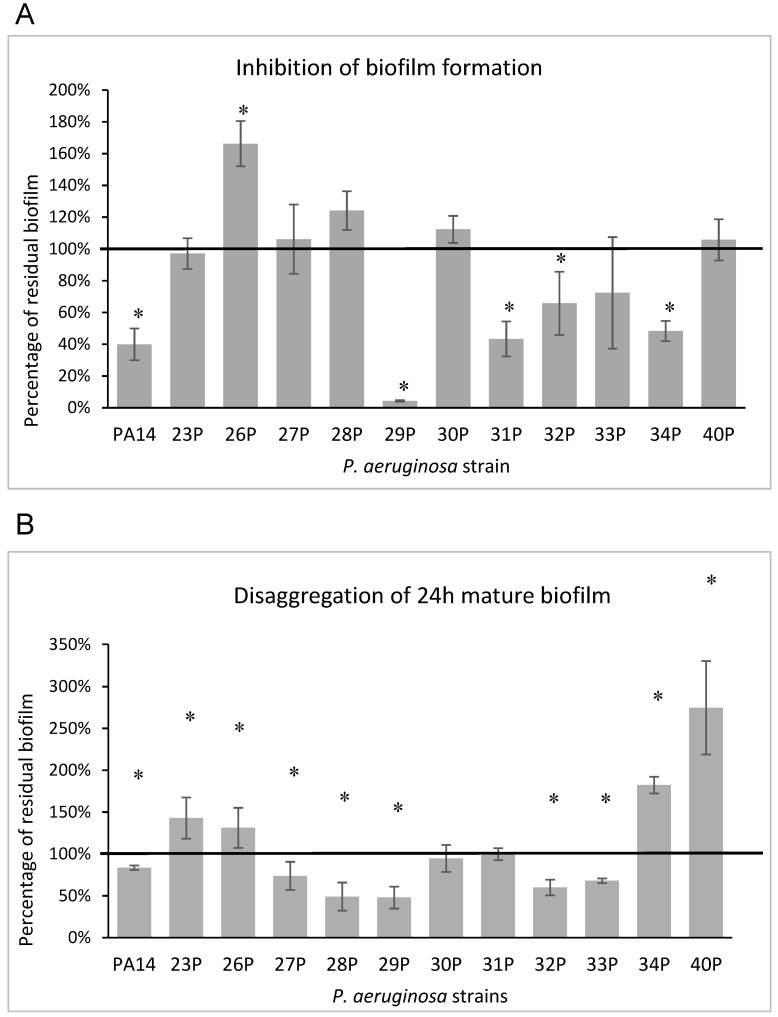
Effect of CCEO on biofilm of different clinical and reference strains at a concentration of 1% *v/v*. Panel (**A**) Effect of CCEO on biofilm formation. The ordinate axis reports the percentage of bacterial biofilm production. Data are expressed as the percentage of biofilm formed in the presence of CCEO compared with the untreated sample. Panel (**B**) Effect of CCEO on 24 h mature biofilm. The ordinate axis reports the percentage of residual biofilm. Data are expressed as the percentage of residual biofilm after 24 h of treatment compared with the untreated sample. Each data point is composed of 4 independent experiments, each performed at least in triplicate. Differences in mean absorbance were compared to the untreated control and considered significant when *p* < 0.05 (* means *p* < 0.05) according to the Student’s *t*-test.

**Figure 2 microorganisms-09-02257-f002:**
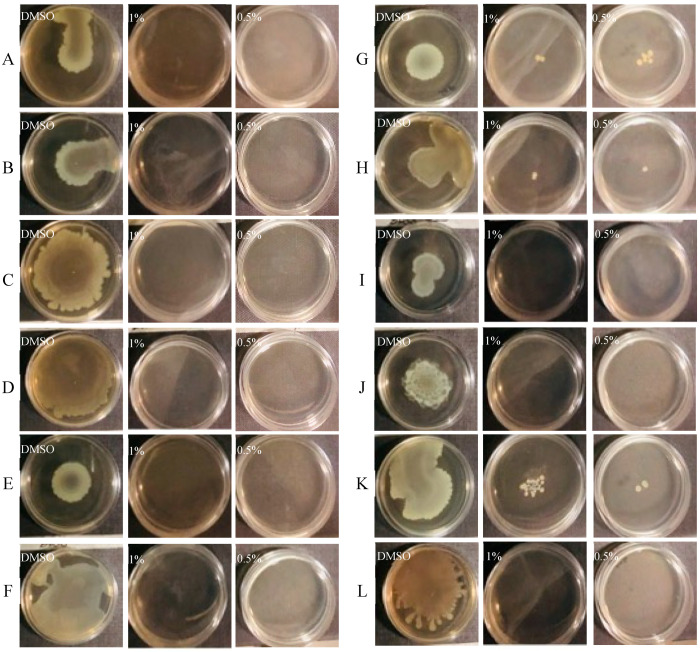
Swarming inhibition assay of CCEO at two different concentrations on reference and clinical strains. (**A**) *P. aeruginosa* PA14; (**B**) *P. aeruginosa* 23P; (**C**) *P. aeruginosa* 26P; (**D**) *P. aeruginosa* 27P; (**E**) *P. aeruginosa* 28P; (**F**) *P. aeruginosa* 29P; (**G**) *P. aeruginosa* 30P; (**H**) *P. aeruginosa* 31P; (**I**) *P. aeruginosa* 32P; (**J**) *P. aeruginosa* 33P; (**K**) *P. aeruginosa* 34P; (**L**) *P. aeruginosa* 40P.

**Figure 3 microorganisms-09-02257-f003:**
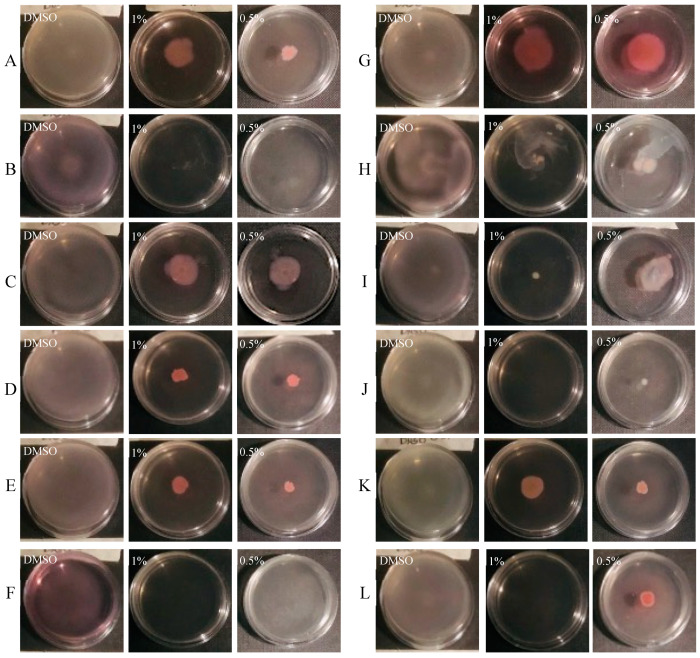
Swimming inhibition assay of CCEO at two different concentrations on reference and clinical strains. (**A**) *P. aeruginosa* PA14; (**B**) *P. aeruginosa* 23P; (**C**) *P. aeruginosa* 26P; (**D**) *P. aeruginosa* 27P; (**E**) *P. aeruginosa* 28P; (**F**) *P. aeruginosa* 29P; (**G**) *P. aeruginosa* 30P; (**H**) *P. aeruginosa* 31P; (**I**) *P. aeruginosa* 32P; (**J**) *P. aeruginosa* 33P; (**K**) *P. aeruginosa* 34P; (**L**) *P. aeruginosa* 40P.

**Figure 4 microorganisms-09-02257-f004:**
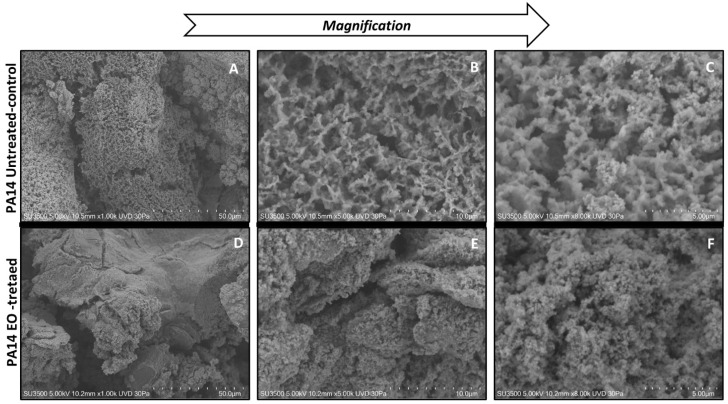
SEM analysis of *P. aeruginosa* PA14 treated and not treated with CCEO. Figures (**A**–**C**) VP-SEM images of untreated *P. aeruginosa* PA14. (**A**) Low magnification (1.00K). (**B**) Medium magnification (5.00K). (**C**) High magnification (8.00K). Figures (**D**–**F**) VP-SEM images of *P. aeruginosa* PA14 treated with CCEO. (**D**) Low magnification (1.00K). (**E**) Medium magnification (5.00K). (**F**) High magnification (8.00K).

**Figure 5 microorganisms-09-02257-f005:**
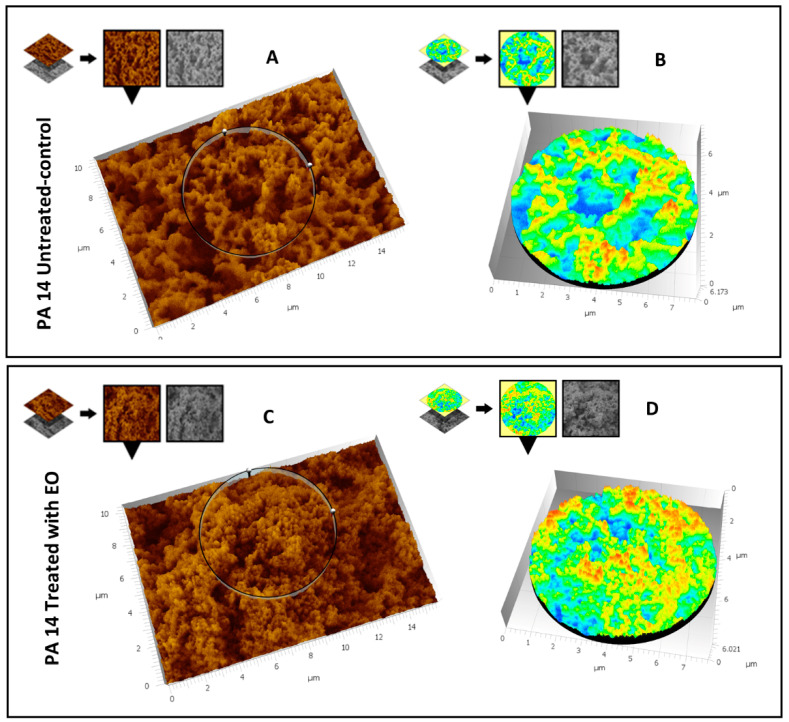
Three-dimensional reconstruction of Figure 4C (untreated sample) and Figure 4F (CCEO treated sample). (**A**,**B**) represent the 3D reconstruction of the untreated sample; (**C**,**D**) illustrate the 3D reconstruction of the treated sample. Lower areas are represented in dark blue; mild and high elevated areas correspond to ECM mesh frames, color varies from yellow and red (high elevated areas). The deep channels areas are represented in blue, and the trabecular system in turquoise-green (mildly elevated areas) and yellow-red (highest areas). (**A**) untreated sample; (**B**) treated sample.

**Figure 6 microorganisms-09-02257-f006:**
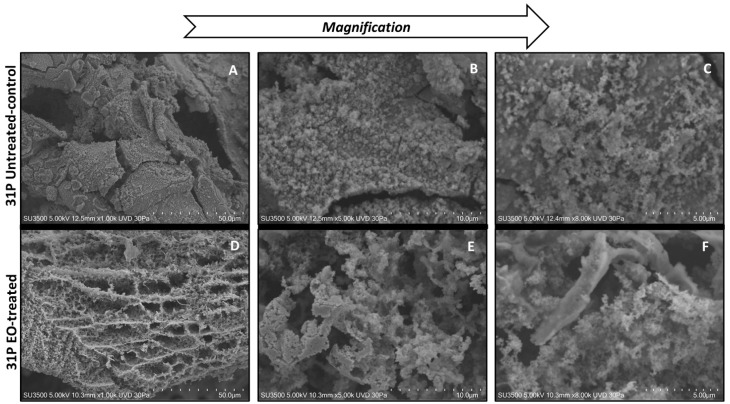
SEM analysis of *P. aeruginosa* 31P treated and not treated with CCEO. Figures (**A**–**C**): VP-SEM images of untreated *P. aeruginosa* 31 P. (**A**) Low magnification (1.00K). (**B**) Medium magnification (5.00K). (**C**) High magnification (8.00K). Figures (**D**–**F**): VP-SEM images of *P. aeruginosa* 31 P treated with CCEO. (**D**) Low magnification (1.00K). (**E**) Medium magnification (5.00K). (**F**) High magnification (8.00K).

**Figure 7 microorganisms-09-02257-f007:**
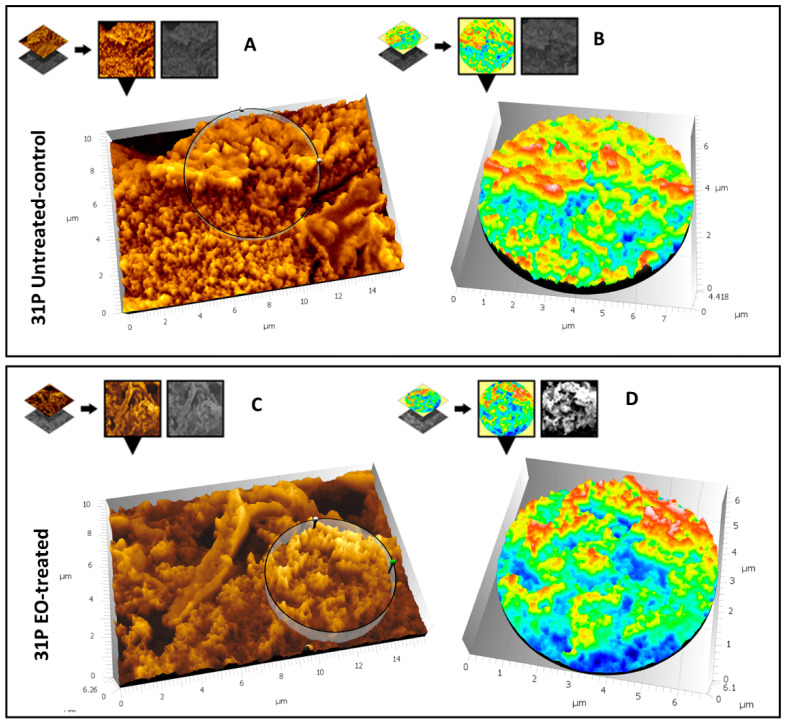
Three-dimensional reconstruction of Figure 6C (untreated sample) and Figure 6F (CCEO treated sample). Lower areas are represented in dark blue; mild-elevated areas are represented in turquoise-green; the color of high-elevated areas varies from yellow to red. (**A**) represents the 3D reconstruction of the untreated sample of Figure 6C. (**B**) is the extracted circular area from (**A**) represented in false colors. (**C**) represents the 3D reconstruction of the treated sample of Figure 6F; (**D**) illustrates the extracted circular area from (**C**).

**Table 1 microorganisms-09-02257-t001:** Chemical composition (%) of CCEO used in this manuscript, obtained by GC-MS [36].

Chemical Component	Peak Area (%)
Carvacrol	61.0
β-caryophyllene	13.6
p-cymene	7.1
γ-terpinene	6.0
Linalool	1.8
β-bisabolene	1.6
β-myrcene	1.6
α-terpinene	1.5
caryophyllene oxide	1.4
α-thujene	1.0
1-octen-3-ol	0.7
Borneol	0.6
Humulene	0.6
Thymol	0.5
Limonene	0.3
β-phellandrene	0.3
cis- β-terpineol	0.2
α-citral	0.2
Total (%)	100

**Table 2 microorganisms-09-02257-t002:** Phenotypic characterization of clinical and PA14 strains.

*Bacterial Strain*	*Pyocyanin* *(OD 520 nm)*	*Biofilm ^a^* *(OD 590 nm)*	*Biofilm ^b^* *(OD 590 nm)*	*Swarming Motility ^c^*	*Swimming Motility ^c^*
PA14	0.178 ± 0.042	3.561 ± 0.357	13.470 ± 1.403	+++	+++
23P	0.148 ± 0.024	3.175 ± 0.851	0.738 ± 0.373	+++	+++
26P	0.120 ± 0.024	0.656 ± 0.281	2.107 ± 1.182	+++	+++
27P	0.102 ± 0.005	1.429 ± 0.643	3.049 ± 0.796	+++	+++
28P	0.093 ± 0.032	0.265 ± 0.038	2.540 ± 0.799	+++	+++
29P	0.199 ± 0.059	0.294 ± 0.066	10.432 ± 3.483	+++	+++
30P	0.115 ± 0.022	0.866 ± 0.345	5.013 ± 1.391	+++	+++
31P	0.063 ± 0.031	1.741 ± 0.154	5.133 ± 0.946	+++	+++
32P	0.121 ± 0.037	1.117 ± 0.163	5.597 ± 1.390	+++	+++
33P	0.050 ± 0.033	0.656 ± 0.115	4.148 ± 1.173	+++	+++
34P	0.094 ± 0.031	1.024 ± 0.212	3.072 ± 0.611	+++	+++
40P	0.160 ± 0.041	0.970 ± 0.201	2.172 ± 0.194	+++	+++

^a^ Biofilm production during an incubation period of 24 h without medium replacement. ^b^ Biofilm production during an incubation period of 48 h with medium replacement after 24 h. ^c^ +++ Motility zone (mm) > 30 mm.

**Table 3 microorganisms-09-02257-t003:** Effect of CCEO on pyocyanin production. Data are reported as the percentage of residual pyocyanin production after the CCEO treatment in comparison with untreated controls. Each data point is composed of 4 independent experiments, each performed at least in triplicate.

Bacterial Strain	Percentage of Pyocyanin Production
PA14	7%
23P	0%
26P	0%
27P	0%
28P	0%
29P	0%
30P	0%
31P	0%
32P	16%
33P	2%
34P	0%
40P	7%

## Data Availability

Not applicable.

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
