# Peer review of "Anti-Virulence Properties of Coridothymus capitatus Essential Oil against Pseudomonas aeruginosa Clinical Isolates from Cystic Fibrosis Patients"

_microorganisms, 2021, doi:10.3390/microorganisms9112257_

Round 1

Reviewer 1 Report

General comments

This study is very interesting in that it tries to address the prevailing problem in CF patients. The study was well-designed and well-written. My general suggestion is margining short paragraphs together in order to improve the readability of this manuscript.

Specific comments

Introduction

The background information and the research problem were well stated. The rationale for this study was that P. aeruginosa forms biofilms; however, this bacterial species is well known for AMR, and would be good mentioning about that in this section. Some of the paragraphs are very short or a single long sentence, I would merge them.

M & M

The study designs were sound scientifically and well described and supported with citations.

Some paragraphs are very short, I would merge them with each other.

What were the number of replicates used in these experiments?

Line 111… “10.000 isolates”, do you mean ten (10.00) or ten thousand (10,000)?

Results

The results are well elaborated and accompanied by well-prepared tables, figures and graphs. Merging the short paragraphs would improve the readability of this manuscript.

Discussion and conclusion

Excellent! The in vitro experiments clearly indicated the benefit of EO in impacting the virulence factors of PA. In this section, its benefit in treating CF patients has been suggested. But in vivo studies are required before trying it in humans. I would discuss about in vivo studies; what would be a good animal model for such studies?

Author Response

Reply to reviewers 1

Reviewer 2 Report

In their study entitled ‘Anti-virulence properties of Coridothymus capitatus essential oil against Pseudomonas aeruginosa clinical isolates from cystic fibrosis patients’ Gianluca et al. investigate the effect of the essential oil (EO) of thyme on a selection of clinical isolates of PA. For this purpose, they use standardized tests that assess the expression of various virulence factors of PA. Particularly to be emphasized is the impressive SEM imaging and its computer-aided 3D evaluation.

Although the manuscript is generally pleasant to read, there are some linguistic deficiencies so I would recommend revision by a native speaker.

The introduction describes the scenario of CF lung disease relatively simple. So, the authors describe the function of the CFTR channel simply as transport, but should at least specify it  as an ion transporter and better go a little bit more into detail. When Gianluca et al. describe the adaptation process of P. aeruginosa to the CF lung, they cite with Winstanley et al. (2016) already a highly important reference, however, I would prefer to see more detail on the evolution of the adaptation of Pseudomonas and therefore recommend further references, e.g. PMID: 29470920 or PMID: 19423715. Moreover, the ductus of the introduction gives the reader the feeling that CCEO is to be investigated with regard to its antibacterial properties, since the authors describe the complicated eradication of bacterial pathogens from the CF lung or that inhibitors of QS could be used as a new treatment option for multi-resistant germs.

At the end, however, CCEO is recommended to be used as an anti-infective agent for surfaces that bear a high risk of being colonised with biofilm-forming PAs. This use is certainly also very substantial, but the authors’ statements in the introduction rather suggest its use as a therapeutic agent for CF therapy and are therefore misleading. Only in the very last paragraph of the discussion the authors describe the possible use of CCEO in the disinfection of respiratory care equipment to prevent or control the spread and transmission of the germ. Although I support the publication of the study, I would recommend a rewording of the introduction and also the title to allow the reader immediately to deduce of what the study is actually about.

Results:

The findings of the two biofilm assays are not consistent for the tested isolates, the authors should at least discuss why this might be the case or – even better- do another assay to verify their results. For example, the pellicle formation assay is quite simple and easy to do (ref. PMID: 31105285). Last but not least the authors would tremendously improve the impact of their ms, if they could additionally provide information on antibiotic resistences of the tested strains and correlate this with the effect of CCEO on the single isolates.

Minor comments:

Page 1, line 39: wrong spelling of CFTR, the human gene must be written with capital letters and italics

Page 2, line 58/59: sentence structure

Page 2, line 88: CF instead of FC! Same on page 15 line 426.

Page 15, line 428: grammar

Author Response

Reply to reviewers 2 

In their study entitled ‘Anti-virulence properties of Coridothymus capitatus essential oil against Pseudomonas aeruginosa clinical isolates from cystic fibrosis patients’ Gianluca et al. investigate the effect of the essential oil (EO) of thyme on a selection of clinical isolates of PA. For this purpose, they use standardized tests that assess the expression of various virulence factors of PA. Particularly to be emphasized is the impressive SEM imaging and its computer-aided 3D evaluation.

Although the manuscript is generally pleasant to read, there are some linguistic deficiencies so I would recommend revision by a native speaker.

We thank the reviewer for these general comments and we amended the manuscript as requested.

The introduction describes the scenario of CF lung disease relatively simple. So, the authors describe the function of the CFTR channel simply as transport, but should at least specify it as an ion transporter and better go a little bit more into detail.

We thank the reviewer for this comment and we added the following period in the Introduction section:

Lanes 39-43

CF is a genetic disorder caused by the presence of mutations in CFTR gene encoding for a protein called cystic fibrosis transmembrane conductance regulator (CFTR), a cell-surface localised chloride channel that regulates absorption and secretion of salt and water across epithelia [3]. 

When Gianluca et al. describe the adaptation process of P. aeruginosa to the CF lung, they cite with Winstanley et al. (2016) already a highly important reference, however, I would prefer to see more detail on the evolution of the adaptation of Pseudomonas and therefore recommend further references, e.g. PMID: 29470920 or PMID: 19423715.

We thank the reviewer for this comment and we added the following period in the Introduction section:

Lanes 59-68

During long-term lung infection in CF patients, P. aeruginosa strains develop mutations leading to clonal expansion. This microevolution is believed to be correlated with a reduced virulence. Various phenotypic traits have been described in P. aeruginosa strains derived by chronic infections, which are absent in environmental P. aeruginosa strains, including loss of motility, acquisition of mucoidy, and antibiotic resistance [Bragonzi, A.; Paroni, M.; Nonis, A.; Cramer, N.; Montanari, S.; Rejman, J.; Di Serio, C.; Döring, G.; Tümmler, B. Pseudomonas aeruginosa microevolution during cystic fibrosis lung infection establishes clones with adapted virulence. Am J Respir Crit Care Med 2009, 180, 138-45].

Indeed, in literature it is reported that microevolution within CF lungs leads to the selection of P. aeruginosa strains with altered but not reduced virulence. This evolution does not do not reflect merely the natural course of infection, since it is the result of the interaction between pathogen, host, and treatment. Furthermore, the genetic adaptation of P. aeruginosa can be distinct in differentially predisposed hosts [Klockgether, J.; Cramer, N.; Fischer, S.; Wiehlmann, L.; Tümmler, B. Long-Term microevolution of Pseudomonas aeruginosa differs between mildly and severely affected cystic fibrosis lungs. Am J Respir Cell Mol Biol 2018, 59, 246-256].

We also added the sentence reported below about AMR of Pseudomonas aeruginosa, as requested by another reviewer.

Lanes 55-58

For these reasons, P. aeruginosa acquired increasing levels of antimicrobial resistance (AMR) in response to treatments. AMR, together with the extensive number of its virulence factors, renders P. aeruginosa an extremely audacious and fearsome pathogen [9].

Moreover, the ductus of the introduction gives the reader the feeling that CCEO is to be investigated with regard to its antibacterial properties, since the authors describe the complicated eradication of bacterial pathogens from the CF lung or that inhibitors of QS could be used as a new treatment option for multi-resistant germs.

At the end, however, CCEO is recommended to be used as an anti-infective agent for surfaces that bear a high risk of being colonised with biofilm-forming PAs. This use is certainly also very substantial, but the authors’ statements in the introduction rather suggest its use as a therapeutic agent for CF therapy and are therefore misleading. Only in the very last paragraph of the discussion the authors describe the possible use of CCEO in the disinfection of respiratory care equipment to prevent or control the spread and transmission of the germ. Although I support the publication of the study, I would recommend a rewording of the introduction and also the title to allow the reader immediately to deduce of what the study is actually about.

We thank the reviewer for this comment and we agree with him/her that this aspect could be misleading.

To better clarify this aspect, we added the sentence reported below in the Introduction section.

Lanes 91-93

Taking into account these features, EOs can be considered promising candidates for both therapy and disinfection of medical devices.

Regarding the title:

“Anti-virulence properties of Coridothymus capitatus essential oil against Pseudomonas aeruginosa clinical isolates from cystic fibrosis patients”

we do not explicitly refer to the use of CCEO in therapy, but to the analysis of CCEO effect of several virulence factors of P. aeruginosa clinical isolated from CF patients. Thus, we do not feel to reword it.

Results:

The findings of the two biofilm assays are not consistent for the tested isolates, the authors should at least discuss why this might be the case or – even better- do another assay to verify their results. For example, the pellicle formation assay is quite simple and easy to do (ref. PMID: 31105285).

We thank the reviewer for this comment but we disagree with him/her.

The biofilm assays we used are universally recognized for the biofilm quantification of clinical and reference isolates, as you can see from some examples of literature reported below and in the manuscript.

  • Pejčić, M.; Stojanović-Radić, Z.; Genčić, M.; Dimitrijević, M.; Radulović, N. Anti-virulence potential of basil and sage essential oils: inhibition of biofilm formation, motility and pyocyanin production of Pseudomonas aeruginosa Food Chem Toxicol 2020, 141, 111431. IMPACT FACTOR: 6.023
  • Papa, R.; Garzoli, S.; Vrenna, G.; Sabatino, M.; Sapienza, F.; Relucenti, M. et al. Essential oils biofilm modulation activity, chemical and machine learning analysis. Application on Staphylococcus aureus isolates from cystic fibrosis patients. Int J Mol Sci 2020, 21, 9258. IMPACT FACTOR: 5.923
  • Kosuru RY, Roy A, Bera S. Antagonistic Roles of Gallates and Ascorbic Acid in Pyomelanin Biosynthesis of Pseudomonas aeruginosa Curr Microbiol. 2021 Nov;78(11):3843-3852. IMPACT FACTOR: 2.188
  • O'Toole GA. Microtiter dish biofilm formation assay. J Vis Exp. 2011 Jan 30;(47):2437. IMPACT FACTOR: 1.4

The reference suggested by the reviewer is referred to:

Samad A, Khan AA, Sajid M, Zahra R. Assessment of biofilm formation by Pseudomonas aeruginosa and hydrodynamic evaluation of microtiter plate assay. J Pak Med Assoc. 2019 May;69(5):666-671. PMID: 31105285. IMPACT FACTOR: 0.781

The aim and scope of this journal reports: “The Journal of Pakistan Medical Association publishes scholarly research focusing on the various fields in the areas of health and medical education. It publishes original research describing recent advances in health particularly clinical studies, clinical trials, assessments of pathogens of diagnostic importance, medical genetics and epidemiological studies”.

Thus, this journal is not specifically focused on microbiology but on a variety of different diverging generic topics.  

Furthermore, in the Materials and Methods section of the suggested manuscript three different assays for biofilm were described, only one dedicated to the quantitative analysis of the biofilm. The quantitative assay is the same used by us.

About the suggested pellicle formation assay, it is only qualitative and we feel it is less performing of the solubilization of crystal violet and the subsequent spectrophotometric determination and quantification.

Last but not least the authors would tremendously improve the impact of their ms, if they could additionally provide information on antibiotic resistances of the tested strains and correlate this with the effect of CCEO on the single isolates.

We thank the reviewer for this comment and we agree with him/her that a correlation between the antimicrobial profile and the effect of CCEO on virulence factors of P. aeruginosa isolates could be very interesting. The antibiotic resistances for each clinical strain are reported, together with other features, in the supplementary materials of the manuscript. However, we sought to find a connection but unfortunately the virulence determinants here investigated do not seem to correlate with the antibiotic resistance profile of each strain. For example, 23P is a multidrug resistant strain while 40P is sensitive to all tested antibiotics, but they have almost the same behavior when treated with CCEO.

Minor comments:

Page 1, line 39: wrong spelling of CFTR, the human gene must be written with capital letters and italics

We apologize for this spelling mistakes. We corrected it.

Page 2, line 73/75: sentence structure

The sentence was reworded as reported below:

In a biofilm, different bacterial communities are incorporated in a self-produced ex-opolysaccharide matrix (EPS), that protects bacteria from antibiotics, ensuring them the survival and complicating their eradication.

Page 2, line 88: CF instead of FC! Same on page 15 line 426.

We apologize for this spelling mistakes. We corrected it.

Page 15, line 428: grammar

We apologize for this spelling mistakes. We corrected it.

Reviewer 3 Report

Dear Authors,

The manuscript ID: microorganisms_1386527 entitled “Anti-virulence properties of Coridothymus capitatus essential oil against Pseudomonas aeruginosa clinical isolates from cystic fibrosis patients” written by Vrenna Gianluca, Artini Marco, Ragno Rino, Relucenti Michela, Fiscarelli Ersilia Vita, Tuccio Guarna Assanti Vanessa, Papa Rosanna and Selan Laura is devoted to specific biological properties of Coridothymus capitatus essential oil (CCEO).

The activity of CCEO against particular virulence factors, produced by P. aeruginosa such as biofilm formation and accumulation, pyocyanin production, and swimming and swarming motility was shown in the present work. This activity is definitely related to its chemical profile rich in phenolic compounds, terpens and terpinenes.

The whole manuscript (Introduction, Materials and Methods, Results, Discussion and References) is properly organized and well written. Introduction contains general data on P. aeruginosa, its pathogenicity and virulence factors. The purpose of the work is concrete. The research carried out using various methods with the involvement of the statistical analysis. The results are well documented, presented in the form of figures (e.g. high-quality photos) and tables and right interpreted. Based on the results an interesting discussion was prepared. Moreover, adequate conclusions were drawn that CCEO could be a candidate to be used for patient’s used equipment disinfection procedures.

I think, it is very interesting and valuable article.

I have only very small suggestions in order to improve paper, which are the following:

  • P. aeruginosa - sometimes it was not italicized, e.g. line 305, 352;
  • The summary of the discussion can be moved to a separate chapter "Conclusions".

In my opinion, this manuscript is worth publishing in “Microorganisms”.

With highest regards,

Author Response

Reply to reviewers 3

Reviewer 4 Report

Anti-virulence properties of Coridothymus capitatus essential oil against Pseudomonas aeruginosa clinical isolates from cystic fibrosis patients.

Manuscript is good written article, which presents the results of research on the operation of Coridothymus capitatus essential oil against Pseudomonas aeruginosa. So far, such studies have not been carried out. I only found minor bugs that need to be corrected.

Throughout the manuscript, the abbreviation CCEO should be used wherever Coridothymus capitatus essential oil is mentioned.

In a lot of Figures and Tables their description is too long.

Abstract

Line 28 – instead of “drastic reduction” should be specific values given.

Materials and Methods

Line 144 – it should be “10^6”

Line 198-199 – “P.aeruginosa” – italic

Line 214 – “0.5% (V/V)”

Results

Line 239 – Table 2 – “c + Motility zone” – add here “+”

Line 242-243 – delete “enter”

Line 253 – In Figure 1 I see only 4 not 5 strains

Line 256 - the closing bracket is missing

Line 324 – delete the dot “Figure 4F”

Line 352 – why authors choose P. aeruginosa 31P?

Author Response

Reply to reviewers 4

Round 2

Reviewer 2 Report

I appreciate the changes made in the manuscript and the response the authors made to my comments. I accept the authors' criticism of the proposed biofilm assay. Although I was aware that suggested test was a qualitative one, I had the feeling that it could further generalize the statement about biofilm formation made in the manuscript. However, I agree with the authors that the journal I referred to does not have a very high impact. I deliberately refrained from suggesting that the authors test their isolates in other in vitro or even in vivo models, because I think this would have gone beyond the scope of the study. In summary, I state that the authors’ adequately answered all my questions and edits, thus I recommend the manuscript for publication in its revised form.